# A Compact, Low-Cost, and Low-Power Turbidity Sensor for Continuous In Situ Stormwater Monitoring

**DOI:** 10.3390/s24123926

**Published:** 2024-06-17

**Authors:** Miao Wang, Baiqian Shi, Stephen Catsamas, Peter Kolotelo, David McCarthy

**Affiliations:** 1Department of Civil Engineering, Monash University, Wellington Road, Melbourne 3800, Australia; miao.wang@monash.edu (M.W.); baiqian.shi@monash.edu (B.S.); stephen.catsamas@monash.edu (S.C.); peter.kolotelo@monash.edu (P.K.); 2School of Civil and Environmental Engineering, Queensland University of Technology, S Block, Level 7, S727, Brisbane 4000, Australia

**Keywords:** sediment, real-time, IoT, urban water, stormwater management, turbidity

## Abstract

Turbidity stands as a crucial indicator for assessing water quality, and while turbidity sensors exist, their high cost prohibits their extensive use. In this paper, we introduce an innovative turbidity sensor, and it is the first low-cost turbidity sensor that is designed specifically for long-term stormwater in-field monitoring. Its low cost (USD 23.50) enables the implementation of high spatial resolution monitoring schemes. The sensor design is available under open hardware and open-source licences, and the 3D-printed sensor housing is free to modify based on different monitoring purposes and ambient conditions. The sensor was tested both in the laboratory and in the field. By testing the sensor in the lab with standard turbidity solutions, the proposed low-cost turbidity sensor demonstrated a strong linear correlation between a low-cost sensor and a commercial hand-held turbidimeter. In the field, the low-cost sensor measurements were statistically significantly correlated to a standard high-cost commercial turbidity sensor. Biofouling and drifting issues were also analysed after the sensors were deployed in the field for more than 6 months, showing that both biofouling and drift occur during monitoring. Nonetheless, in terms of maintenance requirements, the low-cost sensor exhibited similar needs compared to the GreenSpan sensor.

## 1. Introduction

Water quality has significant environmental and public health implications [1,2,3,4]. Addressing this issue requires a comprehensive understanding of pollutant distribution in catchments to enable targeted protective measures [5,6,7,8]. To attain detailed insights, high-resolution pollutant data is essential [9,10] to identify problematic areas and sources, facilitating focused interventions [11]. 

Turbidity, which quantifies water’s optical clarity or cloudiness [12], is a valuable parameter [13,14,15,16,17] that reliably indicates total suspended solids and suspended sediment concentration [18,19]. It can also infer the presence of other pollutants like suspended micro-organisms [20,21,22], making it a significant water quality surrogate. Real-time, high-temporal and spatial resolution turbidity data collection is desired for an accurate understanding of urban water system quality [23].

The conventional method for measuring turbidity is via a manually collected water sample, which is then either read with a hand-held turbidimeter or transported back to the lab for measurement [24]. However, collecting water samples and measuring their turbidity demands significant labour resources, as individuals need to visit different locations to collect samples. Although portable hand-held turbidity meters have been developed for field measurements, they are unable to capture temporal turbidity variations comprehensively, as they only provide turbidity information of the manually collected samples. This limitation makes it challenging to accurately assess pollutant conditions, particularly in stormwater systems where sediment levels are known to have high variability [25]. 

Monitoring turbidity continuously using high-end in-field sensors allows for more comprehensive temporal turbidity data collection. However, these sensors face challenges in capturing high spatial resolution in turbidity due to their high cost (e.g., more than USD 2000 for each GreenSpan TS-1000A turbidity sensor) [26], space requirements (bulky batteries and dataloggers are required for data collection), and installation difficulties (shelter is required for the sensor and a cabinet needs to be installed for the battery and datalogger) [27]. 

In recent years, researchers have developed low-cost devices for measuring turbidity in surface water, drinking water, or natural water bodies [28,29,30,31,32,33,34,35,36,37,38,39,40,41]. Although these developments demonstrated good correlation between the measured results and the reference turbidity readings, many of them exhibit limitations when it comes to meeting the requirements for in situ, field monitoring applications. Several designs lack robust waterproof enclosures, making them unsuitable for practical field deployments [28,31,34,37,38,41]. For instance, Wang et al. (2018) designed a turbidity sensor for fresh water monitoring [41], but the device is primarily an experimental setup involving connected electrical components without a waterproof enclosure.

Other designs require water samples to be placed in bottles or containers for turbidity measurements [32,35,39,40], limiting their capability for continuous monitoring. One instance that can be found is that Kitchener et al. (2019) developed a sensor that requires leaving water samples in a cylinder for turbidity testing [40], making it unsuitable for long-term field deployment.

Only a few designs have the capability to directly measure water turbidity in the water column without the need for sampling processes [29,30,33,36]. However, these sensors impose constraints such as channelling the water through a specific tubing or extracting water into a chamber or container. For example, Azman et al. (2017) required water to be passed through a pipe to enable accurate measurements, thus imposing tight constraints on the scenarios where this sensor can be used [30].

Notably, the aforementioned designs were primarily tested in lab environments, lacking comprehensive in situ validation. As sensor performance could be affected by natural factors (sunlight, biofilm, etc.), resulting in lower accuracy and reliability [42], the in situ performance of these sensors requires thorough investigation.

There were two occasions where long-term applicability was considered in low-cost turbidity sensor development. However, the focus frequently shifts to different water domains, limiting its direct application to stormwater systems (such as drainage networks or flowing water channels). For example, Oscar et al. (2020) designed a turbidity meter (a disk floating on the stable water surface) capable of transmitting monitoring results and GPS information via satellite communication [43], yet the design is focused on open-water body/lake monitoring and is unsuitable for monitoring flowing stormwater drains. Similar challenges apply to the sensor designed by He et al. [44] (2020), and although the sensor boasts high sensitivity, its bulky size prevents its application in urban stormwater systems. 

Based on our literature review, there is a lack of research specifically on in situ and continuous turbidity sensors, tailored to the unique requirements and challenges of extended field deployments in urban water systems. As such, this paper presents the design and both the laboratory and the field validation of a low-cost turbidity sensor suitable for long-term implementation specifically in urban water systems. This sensor is an assembled waterproof product, allowing for continuous turbidity monitoring over an extended period and exposure to the complex natural environment. Moreover, its compact size and the mounting option enable easy deployment in different field conditions with minimal setup time.

Both lab calibration and field validation were conducted for testing the sensor’s performance. After testing its laboratory performance, its field application and validation were conducted in Troups Creek Wetland, located in Narre Warren North, southeastern Melbourne. The objective of this work was to verify whether this innovative technology could meet the following requirements: (1) provide reliable, long-term monitoring results compared to a high-end turbidity sensor (GreenSpan TS-1000A turbidity sensor); (2) be of low cost; (3) have low power consumption; (4) be resistant to biofouling; and (5) have minimal drift issues after long-term deployment.

## 2. Materials and Methods

### 2.1. Turbidity Measurement Mechanism

The mechanism used for water turbidity measurement in the proposed sensor relies on optical-based sensing techniques [45]. This method utilises reflectance principles [19], wherein light emitted from a source is reflected by particles present in the water before being detected by a photodetector. The amount of reflected scattered light received by the photodetector increases with the presence of more particles in the water column.

The selection of the key components and the sensor design follow the Standard ISO7027 [46], which indicates the light source should be near-infrared with a wavelength of 860 ± 30 nm to minimize the interference from coloured samples. Therefore, the light emitter and the photodetector of the low-cost innovative sensor are selected as an 850 nm infrared LED (VSLY5850 LED) and an 850 nm infrared phototransistor (BPW77NB), respectively. The resistance of the phototransistor is light-dependent, leading to a decrease in resistance as it detects higher levels of light (indicating increased turbidity) and an increase in resistance under low-light conditions (indicating low turbidity). To facilitate this measurement, the phototransistor is integrated into a voltage divider configuration in series with a resistor. An ATmega328P microcontroller (MCU) is employed to gauge the voltage drop across the resistor within the circuit using an analogue pin. Consequently, a positive correlation between turbidity and the voltage drop emerges, allowing the creation of a calibration curve based on this relationship. 

### 2.2. Electrical and Physical Overview

The mechanism used for water turbidity measurement in the proposed sensor relies on optical-based sensing techniques [45]. This method utilises reflectance principles [19], wherein light emitted from a source is reflected by particles present in the water before being detected by a photodetector. The amount of reflected scattered light received by the photodetector increases with the presence of more particles in the water column.

The turbidity sensor is able to operate under a 3.3 V power supply and communicate via a universal asynchronous receiver-transmitter (UART). The sensor circuit, as depicted in Figure 1a, utilises an ATmega328P chip as the MCU due to its ease of use and familiarity in sensing applications [47,48]. The circuit includes one LED, one phototransistor, and a voltage regulator to match the nominal voltage of the infrared LED (1.65 V [49]). The phototransistor is powered by an NPN transistor. A 2 MΩ resistor forms a voltage divider with the phototransistor with the current range from 0 mA to 0.00165 mA, and the MCU measures the voltage drop across the resistor; this voltage drop is converted to digital values by the MCU’s 10-bit analogue to digital converter (ADC) such that the output of the sensor is an integer between 0 and 1023. These integers are positively correlated with turbidity. The details of the drawings and the design files are uploaded to the Appendix A, making it easy for readers to check, use, and improve. 

During the sensor development process, more powerful electrical components were considered to potentially provide monitoring results with higher accuracy. However, factoring in cost constraints and the specific monitoring requirements, the current component selections were made as they satisfactorily meet the necessary criteria. For instance, the ATmega328P microcontroller, despite being a 10-bit system, can provide 1024 distinct digital values. Given that the targeted turbidity measurement range is 0–250 NTU for stormwater [50,51], this resolution (where 1 bit indicates 0.25 NTU) offers sufficient accuracy for a low-cost device. Moreover, non-polar capacitors were utilised as they could adequately serve the intended function while being more cost-effective for the low capacitance values required.

To ensure the sensor can be easily assembled without complicated wire connection, a printed circuit board (PCB) (Figure 1b) was designed. The PCB has a compact dimension of 24 mm by 12 mm, with its smaller size allowing the sensor to fit into small housings, thereby increasing its applicability across a broader range of applications. To ensure reliable performance in the urban water system, a 3D printed housing is utilised to accommodate the PCB and components. Within the housing, the LED and phototransistor are fixed facing outwards at a 90-degree angle to each other and spaced apart by 22 mm (centre to centre). A 3D printed LED and phototransistor holder is applied to arrange the LED and phototransistor at fixed positions angled at 90-degrees to one another. A transparent epoxy board is placed in front of the LED and phototransistor (Figure 1c) to provide protection. The 3D printed housing is filled with a potting compound to ensure water resistance [52]. The design files of the sensor are available at Mendeley data in [Turbidity sensor design file] at [10.17632/6pcs9hprws.1], published on 29 May 2024.

### 2.3. Removal of Ambient Interference

Ambient light sources, such as sunlight, introduce interference to the sensor readings because they contain infrared radiation at 850 nm, which can be detected by the phototransistor. As a result, the readings are affected by this interference. To address the issue of ambient noise, particularly sunlight interference, a 3D-printed sensor cover is applied to the turbidity sensor. This cover ensures that the phototransistor only detects light emitted from the LED, effectively eliminating external light sources. Additionally, the cover prevents sunlight penetration, significantly reducing the growth rate of biofilms [53] and minimising the need for maintenance.

When selecting the material for the cover, testing is conducted to ensure that no light can penetrate. For this specific design, different brands of two materials, Polylactic acid (PLA) and polyethylene terephthalate glycol (PET-G), are carefully tested to determine the most effective one. It is important to note that certain materials, even with 100% printing infill, may still allow light to pass through and impact the sensor readings. In the final design, the sensor cover is produced using Bilby 3D (PET-G) material [54].

After applying a cover to the sensor, the readings remained stable, unaffected by sunlight or weather conditions. Furthermore, the daytime and nighttime readings exhibited similar results, indicating that the cover effectively mitigated ambient light interference.

While the cover eliminates most light sources, there may still be a small amount of natural light detected by the phototransistor, causing interference. To mitigate the impact of background noise and achieve more accurate monitoring results, a scanning process is performed twice: once with the LED turned on and once with the LED turned off. The two scanning processes are conducted within two seconds of each other. The reading obtained with the LED turned off serves as the ambient noise, and the final turbidity measurement results are estimated by calculating the difference between the readings obtained when the LED is turned off and on [43].
(1)$$M=R_{on}-R_{off}$$
where *M* is the analogue difference of the sensor readings, *R_on_* is the turbidity sensor reading with LED turned on, and *R_off_* is the turbidity sensor reading with LED turned off.

### 2.4. Sensor Operation

During the operational phase of the turbidity sensor, two types of readings are obtained: the “LED on” and the “LED off”. Each measurement takes one millisecond, and 1000 measurements can be taken during a one-second-long scan. To statistically reduce the variability to acceptable levels in the monitoring activities, especially in cases where the monitoring conditions experience sudden changes, the program employs a method of taking 1000 measurements. The results obtained from these 1000 measurements are then averaged to provide the final monitoring results so that high quality and accurate readings can be obtained.

The turbidity sensor operates in two modes: working mode and sleep mode. In the working mode, the sensor is active, and both the LED and phototransistor consume power to take readings. This mode is utilised when actively monitoring and collecting turbidity data. The sleep mode is designed to conserve energy during long-term continuous field monitoring. In this mode, the device remains active but the electrical components (LED, phototransistor and AT-mega 328 chip) are not powered, resulting in reduced power consumption. Despite being in a low-power state, the sensor can quickly resume active measurements when required. By switching between these modes, the sensor optimizes energy usage while maintaining its functionality for extended monitoring periods.

### 2.5. Sensor Cost

The cost of a single low-cost turbidity sensor amounts to approximately 23.50 USD. The price of each component is listed in Table 1; for the details of the BoM (Bills of Materials), please refer to the Appendix A. The cost breakdown of each component indicates that this sensor is truly a low-cost product, making it widely accessible for anyone to produce. The affordable nature of the components reinforces the sensor’s potential for widespread adoption and accessibility.

### 2.6. Power Consumption

To determine the power consumption of the innovative turbidity sensor, the current of the sensor is measured in both the working and the sleeping modes. This is accomplished by connecting an ammeter in series with the sensor circuit. For the working mode, there are two phases, one with LED on and one with LED off, the measurements of both phases were conducted. In the sleep mode, the sensor transfers to a low power state. The ammeter readings are recorded for both the working mode and the sleep mode, providing data on the power consumption of the sensor in different operating states.

### 2.7. Labortory Calibration

In order to establish the correlation between the sensor’s output (i.e., an integer between 1 and 1023) and turbidity, it is necessary to generate a calibration curve for the sensor. Since we were unsure whether turbidity sensors would vary between individual units, each sensor needed to undergo individual testing to generate its own calibration curve. Since the sensor is designed for general stormwater monitoring, such as wetland or road rainfall runoff, water turbidity is generally lower than 250 NTU in most instances [50,51]. Consequently, the calibration range of 0–250 NTU was selected. During the calibration process, a standard solution (Australian Chemical Reagent, ACR 4000NTU standard turbidity solution) was used as a standard calibrator to create different turbidity solutions; we created roughly 25, 50, 100, 150, and 250 NTU solutions, respectively, and checked them using a commercial Thermo Scientific AQ4500 hand-held turbidimeter (with the resolution up to 0.01 NTU and ±2% of accuracy between 0–250 NTU), which we consider is capable of providing accurate turbidity readings. The turbidity levels of the dosed solutions were evaluated instead of directly employing the standard calibrator solutions. This approach was adopted due to the potential degradation of the standard calibrator solutions during transportation and storage, which could compromise their accuracy and reliability as reference standards. To make sure the hand-held turbidimeter can provide reliable results, the calibration process was conducted by strictly following the user guide [55]. The analogue readings obtained from the low-cost turbidity sensor are compared to the measured turbidity values to assess if a reasonable regression relationship exists. This relationship can then be utilised as a calibration curve for subsequent turbidity measurements. Each solution is tested three times using the low-cost sensor and the hand-held turbidimeter. The average of these three readings is employed for calibration analysis. The expected relationship between the turbidity sensor readings and the hand-held turbidimeter reading values is theoretically linear [56] and can be expressed as:(2)T=aM+b
where *T* is the calibrated turbidity value, *a* is the parameter to indicate the slope of the linear relationship, *M* is the analogue difference of the sensor readings, and *b* is the parameter to indicate the intercept of the linear relationship.

### 2.8. Field Validation

#### 2.8.1. Validation Sites

The low-cost turbidity sensors were validated through a field test at Troups Creek Wetland in Narre Warren, southeast Melbourne. Two locations, the wetland inlet and outlet, were chosen for the sensor installation. These locations represent key points for assessing the removal of turbidity through the wetland, and is commonly done in the literature [57,58,59,60]. The sensors were installed alongside high-end turbidity sensors (GreenSpan TS1000A turbidity sensor, with the resolution up to 0.1 NTU and ±1% between 0 and 250 NTU) and a distance of approximately 30 cm between them. However, due to the installation restrictions, the GreenSpan sensor probe is facing the downstream of the flow while the low-cost sensor is facing across the flow. This setup allowed for monitoring turbidity under similar conditions.

#### 2.8.2. Sensor Installation

To gain a more comprehensive understanding of the site conditions, a low-cost depth, electrical conductivity (EC), and temperature sensor [52] was also installed alongside the low-cost turbidity sensor. To facilitate the installation, both the low-cost turbidity sensor and the depth, EC, and temperature sensor were mounted on a 50 mm PVC pipe using a specially designed, 3D-printed sensor holder (Figure 2a). A logger box, housing an ATmega based data logger (BoSL Board v0.4.1 [61]) and a 3.3 V battery, was securely attached to the top of the PVC pipe (Figure 2b). The logger box featured a drilled hole at its base, allowing the sensor cables to pass through and connect to the logger. This integrated setup, encompassing all the sensors and the logger box, constituted a cost-effective sensor package capable of monitoring water turbidity, EC, temperature, and depth (Figure 2c).

During the installation of the low-cost sensor package, a star picket was previously driven into the wetland base. Subsequently, the PVC pipe was slid over the star picket, providing stability and allowing the unit to be positioned within the wetland for data collection.

#### 2.8.3. Monitoring Regime

Both the low-cost sensor package and the GreenSpan turbidity sensor followed an identical monitoring regime, ensuring consistency in data collection. For both sensors, the scan interval was set at 1 min and the averaging interval and logging intervals were set at 6 min; this means the logger averaged the six scan results (taken at 1 min intervals) and then logged this value (to a remote cloud store for the low-cost sensor system and to a local Campbell Scientific logger for the GreenSpan sensor). The only difference between the two sensors is that the GreenSpan sensor only scans once every minute while the low-cost sensor took the 1000 scanning average for the minutely reading.

#### 2.8.4. Sensor Maintenance and Calibration

To ensure the accuracy of the monitoring results and mitigate the potential impact of biofilm and algae, regular maintenance was carried out on the sensors. The frequency of maintenance was determined based on the sensor’s functionality and the local environmental conditions [62]. In this particular case study, we aimed to conduct maintenance every two weeks. The maintenance process consisted of three steps: before cleaning calibration, sensor cleaning, and after cleaning calibration. Over time, as the sensors remained installed in the wetland, a biofilm developed on the surface of the probe, potentially affecting the sensor readings. Hence, it was crucial to establish calibration curves for both the following conditions: with biofilm (before-cleaning) and without biofilm (after-cleaning). By having these calibration curves, the data could be adjusted appropriately by using the respective calibration before and after the cleaning process. This approach ensured accurate and reliable data analysis and interpretation.

Step 1: Before cleaning. We used the diluted standard turbidity solutions (25, 50, 100, 150, and 250 NTU) for calibration; the preparation methods were the same as the lab calibration process (dilute the 4000 NTU standard turbidity solution). With the sensor becoming dirty after a period of field installation, such as mud or algae settling on the surface, it needed a general clean to avoid contamination of the turbidity solution. When cleaning the sensor, only the sensor body was cleaned, and the sensor surface where the LED and PT transmit/read remained untouched, so the biofilm impact of the sensor was able to be captured. When recording the monitoring data, the sensor probe was submerged in the turbidity solution, the reading of the sensors was taken and recorded, first, then checked against the turbidimeter to test the actual turbidity of the solution, which makes sure that an identical turbidity reading is captured. Both the low-cost sensor and the turbidity meter take three continuous readings and the average is calculated for comparison.Step 2: Cleaning. The probe of the sensor was carefully cleaned by DI water and delicate task wipers (Kimwipes); the sensor probe was wiped gently multiple times until no dirt or biofilm was obvious on the wiper.Step 3: After cleaning. After the probe was cleaned, the checking process was repeated as per Step 1, exactly, to assess after cleaning conditions.

The maintenance procedure described above was implemented for the fixed GreenSpan sensors in the field. To simplify the calibration process for the low-cost sensors, each monitoring site is equipped with two sensors that are periodically swapped to collect comprehensive data throughout the monitoring period. The low-cost sensor calibration process was conducted in the lab, but following the same process as outlined above.

#### 2.8.5. Data Combination and Adjustments Based on Calibration Curves

For each monitoring site, two low-cost sensor packages were utilised, enabling continuous data collection. When one sensor package was removed for maintenance, another package was deployed to ensure uninterrupted monitoring. To mitigate the effects of resuspension attributed to maintenance procedures for both the GreenSpan sensor and the low-cost sensor, the duration of maintenance activities (including the time required for swapping the low-cost sensor in the wetland) was recorded. Subsequently, data collected during these maintenance intervals were excluded from the comprehensive dataset. 

The collected data from the sensors were in raw form and required adjustment using ‘before-cleaning’ and ‘after-cleaning’ calibration curves, as each sensor had two calibration curves for each monitoring period. These curves accounted for the difference in data between conditions with and without biofilm. The monitoring results from each site were consolidated to form a comprehensive database during data analysis. This approach facilitated accurate calibration and integration of the monitoring data for further analysis.

During each monitoring period, the calibration curve exhibited a distinct difference influenced by the presence of biofilm. It was assumed that the biofilm growth rate remained constant over time. The change rate of the calibration curve displayed a linear relationship. Both parameter *a* and parameter *b* in Equation (2) were presumed to change at a constant rate based on the sensor’s in-water duration. As a result, the calibration parameters varied for each monitoring timestamp, and the calibrated turbidity was determined using the specific calibration information available at that timestamp. This approach accounted for the evolving biofilm conditions over time, resulting in more accurate estimations of turbidity and enhancing the reliability of the monitoring data analysis.

#### 2.8.6. Data Validation

The combined and adjusted data underwent a validation process consisting of several criteria. These criteria were derived from available information encompassing the data themselves, maintenance records, environmental factors, and any issues encountered during the measurement process. Additionally, a combination of these three elements was considered [62]. The designed criteria aimed to filter and retain valid data points for subsequent analysis.

Criterion 1: Sensor monitoring status (in-water, for monitoring, or not). The sensor monitoring status was used to determine if the sensor was immersed in the water for monitoring purposes. This test verified whether the sensor remained fully submerged in the water, ensuring reliable monitoring of turbidity. If the water depth was insufficient, such that it fell below the top surface of the turbidity sensor, the collected data were identified as invalid. The sensor could be out of water for calibration and checking, for instance.Criterion 2: Missing data. Missing data at specific timestamps occurred due to various factors such as battery issues, hardware malfunctions, or software problems. In these cases, when the sensor failed to collect data, the corresponding data points at these timestamps were considered invalid or missing.Criterion 3: Turbidity (inside or outside the calibrated range of the sensor). After applying the calibration curves to the raw data from the two sensors, the calibrated turbidity values should fall within the calibrated range. Since the turbidity solutions used for calibration ranged from 0 NTU to 250 NTU, the reliable detection range for both sensors was set within the same range (0–250 NTU). Therefore, any calibrated turbidity values that fell outside this reliable detection range for their respective sensors were identified as not valid.Criterion 4: Continuous trend data. If the monitoring data exhibit a continuous trend of either increasing or decreasing for a period exceeding 7 days, and this trend remains consistent regardless of weather changes, the entirety of the continuous trend data is considered invalid and is assumed to have been caused by rapid build-up of material on the sensors surface.Criterion 5: Significant fouling. When conducting maintenance, if the presence of dirt, sediments, algae, or snails was observed on the surface of the sensor, it could have a substantial impact on the sensor readings. As it is difficult to determine precisely when the dirt started to accumulate on the sensor, the data collected during the monitoring period between the last maintenance and the current maintenance were regarded as uncertain.Criterion 6: Duration after the last maintenance. If the sensor had not undergone maintenance for a period exceeding two weeks, the data collected beyond the two-week mark from the last maintenance were designated as uncertain.Criterion 7: Filtering erratic values. To filter out erratic increases or decreases in sensor data, as well as unrealistic gradients that do not align with physical processes and local environmental conditions, the Page-Hinckley test was applied [63,64]. This testing method involves comparing the absolute sum of the difference between the residue and the cumulative average with a threshold. Determining the appropriate threshold involves an iterative process with a moving window. The moving average and threshold values need to be set differently for each sensor, considering their specific characteristics. Since the residue of turbidity results follows a normal distribution, around 10% of the total data can be expected to be removed based on this criterion. A sensitivity matrix can be constructed for each sensor, illustrating the amount of data to be removed with different moving window sizes and thresholds. This matrix enables the selection of the optimal combination of moving window and threshold values to effectively filter out inconsistent or erroneous data points.

To ensure the presentation of real data, the aforementioned data validation criteria were strictly followed. By adhering to these criteria, only valid data points were retained for further comparison and analysis, maintaining the integrity of the dataset. The percentage of data points removed as a result of this validation process was determined and will be discussed.

#### 2.8.7. Time Series Data Comparison

To evaluate the reliability of the innovative low-cost turbidity sensors, a comparison was made between their long-term continuous monitoring results and the GreenSpan monitoring results. Only the valid data points from both sensors were considered for this analysis. The turbidity changing patterns during the monitoring period were illustrated by plotting time series data for both sensors at each monitoring site. The difference of the monitoring results from the two sensors was determined; the percentage of the absolute 20 NTU difference was also examined. To provide further context, rainfall data were incorporated as turbidity changes are often associated with rainfall events. Additionally, the influence of high-speed wind causing resuspension and increased turbidity was explored by including wind speed data. Other weather data obtained from the Bureau of Meteorology (BoM) were utilised to enhance the understanding and interpretation of the sensor data.

#### 2.8.8. Statistical Analysis of the Comparison between the Two Sensors

A comparison between the data obtained from both sensors was conducted, and statistical analyses were performed to assess the correlation between the two sensors. To quantitatively assess the correlation between the two sensors’ data, a Pearson correlation test [65] was applied to the collected data. The Pearson correlation test was used to evaluate the linear association between the two datasets. As Pearson test is an assessment for linear relationships that could show how much similarity between the two sensor results (how far from the identity line). By reporting the Pearson correlation coefficient and its statistical significance (*p*-value), the quantitative evidence could be provided to support the validity and performance of the low-cost turbidity sensor in comparison to the established commercial sensor.

#### 2.8.9. Biofouling Impact of the Sensors

As biofilm increases on the probe’s surface, the sensor reading may be higher due to increased light reflection by the biofilm. To assess the influence of biofouling, a comparison was made using the calibration information. For each monitoring period, the after-cleaning calibration curve (representing a cleaned sensor) was compared with the before-cleaning calibration curve (representing a sensor potentially affected by biofilm). In this comparison, different raw readings were assigned to the sensors to align their actual turbidity readings with the 25th, 50th, and 75th percentiles of the entire monitoring results. For instance, once the 25th percentile of the raw readings was obtained, this value was applied to both the “before-cleaning” and “after-cleaning” calibration curves to determine the calibrated turbidity results. Then the relative difference between these calibrated results was calculated, facilitating further analytical exploration. This methodology was consistently applied to the 50th and 75th percentiles of the raw readings to obtain a better understanding of the effects of biofilm accumulation over time.
(3)$$R_{bio}=\frac{T_{A}-T_{B}}{T_{A}}$$
where *R_bio_* is the relative difference between ‘after-cleaning turbidity’ and ‘before-cleaning turbidity’. *T_A_* is the ‘after-cleaning turbidity’ reading for one specific monitoring period and *T_B_* is the ‘before-cleaning turbidity’ reading for one specific monitoring period.

To assess the potential impact of biofouling on turbidity readings, the relationship between the relative difference of before-cleaning and after-cleaning readings and the in-water time was examined. This analysis aimed to evaluate whether biofilm growth influenced the turbidity measurements. This analysis was repeated for all cleaning events for both the low-cost sensors and the GreenSpan sensor, and they were then compared. A Wilcoxon Rank Sum test [66] was conducted to assess the difference between the low-cost turbidity sensor and the GreenSpan sensor in terms of biofouling issues. A *p*-value of 0.05 was considered as the acceptable threshold for statistical significance in the Wilcoxon Rank Sum test.

#### 2.8.10. Permanent Drifting of Sensors

Permanent drift refers to the phenomenon where a sensor’s output progressively deviates from its initial value over time. In the case of the turbidity sensor, such drift could be attributed to scratches incurred from prolonged exposure to harsh environmental conditions. To assess the presence of drift issues, it is crucial to compare the sensor’s performance under conditions indicative of optimal functionality, specifically, when the sensor is thoroughly cleaned and unaffected by biofilm accumulation. Therefore, an analysis was conducted by comparing the calibrated results from the “after-cleaning” calibration curves. The analysis of the sensor drift issue parallels that of assessing the impact of biofouling, with the sole distinction being the basis of comparison. Specifically, the relative difference is calculated using the results derived from the ‘after-cleaning’ calibration data. The after-cleaning calibration curves (i.e., from Step 3, above) from these two cleaning instances were compared to assess the following two aspects:Correlation between in-water time and relative difference. The relationship between the in-water time and the relative difference between the calibration curves was examined.Bias after deployment. The comparison of the after-cleaning calibration curves aimed to identify any significant bias or offset that may have occured in the sensors’ readings after being deployed in the water.

By examining these aspects, the presence of permanent drift issues and any potential correlations or biases in the sensor readings after being deployed in water could be evaluated. The relative difference can be calculated as:(4)$$R_{dri}=\frac{T_{B}^{n}-T_{A}^{n}}{T_{B}^{n}}$$
where *R_dri_* is the relative difference between ‘after cleaning’ turbidity and ‘before cleaning’ turbidity. $T_{B}^{n}$ is the ‘after-cleaning’ turbidity reading before the sensor was deployed) for a specific (nth) monitoring period and $T_{A}^{n}$ is the ‘after-cleaning’ turbidity reading after the sensor was retrieved for one specific (nth) monitoring period.

To evaluate the presence of permanent drift issues, this calculation was done for each monitoring period and for each sensor. The summarized data were then subjected to a Wilcoxon Rank Sum test to assess any significant differences between the low-cost and GreenSpan sensors regarding drifting issues.

## 3. Results and Discussion

### 3.1. Power Consumption

In the sensor’s working mode, the LED operates at two different current levels: 88 mA during the active phase and 4 mA during the inactive phase. During its sleep phase, the current draw is exceptionally low, to the extent that it falls to less than 0.1 µA. Based on the current consumption data outlined in Table 2, assuming a measurement frequency of one measurement per minute, the projected total power consumption over a year of continuous monitoring amounts to approximately 13,500 mAh. This means that a single typical 18,650 Li-ion cell with a capacity of 3500 mAh can sustain the turbidity sensor’s continuous operation for approximately three months. It is worth noting that the monitoring duration can be extended by reducing the scan frequency, as a lower scan frequency would increase the sensor’s operational lifespan.

### 3.2. Laboratory Calibration

Our results demonstrate significant linear relationships between the known turbidity levels and the corresponding sensor output obtained from the sensors (Figure 3; R^2^ > 0.99, *p* < 0.01). In this study, a total of 23 sensors were subjected to testing and calibration in the laboratory and Figure 3 showcases three randomly selected example sensors (all sensors shown in Appendix B). However, it is worth noting that each tested sensor exhibits distinct lines of best fit parameters. The slopes of the calibration curves range from 159 to 1770, while the intercepts vary from −104 to 2.41. These variations stem from differences in sensor manufacturing, such as slight disparities in the gap between the LED and the front epoxy board, which ultimately influence the calibration information and suggest that each sensor should undergo calibration checks before deployment.

### 3.3. Field Validation

#### 3.3.1. Data Cleaning

Based on the data cleaning criteria, no data were removed due to Criterion 1 (submersion of sensors), while Criterion 2 (missing data) resulted in data loss primarily caused by battery outage during the extensive COVID-19 lockdowns that occurred in Melbourne during the testing period. The low-cost sensor had more data removed due to Criterion 3 than the GreenSpan did, which could be explained by debris accumulation due to the untimely maintenance during the COVID-19 lockdown. At the outlet, the low-cost sensor and the GreenSpan sensor had 3.4% and 1.5% of data removed, respectively. Delayed maintenance during the COVID-19 outbreak led to approximately 20% of data removal based on Criterion 6. Overall, around 55% of the data was removed, with delayed maintenance accounting for the largest portion of data loss (Table 3). Overall, there was not a big difference between the percentages of total removed data from the GreenSpan and the low-cost sensor.

#### 3.3.2. Time Series Data Comparison

There is a strong positive relationship between rainfall and turbidity, indicating that turbidity rises are consistently associated with rainfall events (Figure 4). The connection can be attributed to stormwater surface runoff, which transports sediments into the wetland, and the high flow that induces resuspension of sediment at the wetland bed [67]. Additionally, it is worth noting that the rising turbidity can also be influenced by the resuspension caused by wind and wave action [68] (the wind data and other weather data can be seen in Appendix C). This can be observed in the inlet data on 30 June, which show that not much rainfall was recorded but higher wind speeds were present. 

The comparison between the turbidity data collected by the GreenSpan and those collected by the low-cost turbidity sensors reveals a generally similar trend and a good response to weather changes. However, some differences are observed during specific periods in the monitoring plots. In the inlet (Figure 4, subplot 1), the GreenSpan sensor tends to exhibit high turbidity readings mainly during rainfall events or periods of high wind speed, whereas the scatter points from the low-cost sensor often appear after rainfall events. The discrepancy is due to two main factors. Firstly, the uncovered GreenSpan sensor is more affected by water flow debris, unlike the low-cost sensor, whose cover reduces debris impact. Secondly, the low-cost sensor’s cover may trap debris and aquatic animals (Appendix D), leading to higher turbidity readings after high flow events. This also affects the data, with the GreenSpan sensor showing more variability. Sensor placement near the weir, where water depth increases, also contributes to the low-cost sensor’s more stable results during rainfall-induced vortex turbulence. 

In addition to comparing the two sensors’ monitoring results with the weather changes, the differences between their readings were also examined (Figure 4, subplot 2 and 4). The analysis revealed that most of the differences fall within ±20 NTU, approximately 97% of the time for the inlet readings and 73% for the outlet readings. However, scattered data points were observed in the collected data. This disparity can be attributed to the inherent monitoring errors and uncertainties associated with field deployments in unstable environmental conditions. Sensors tend to exhibit greater deviations from their technical specifications [42], which are typically determined under controlled laboratory settings, when operated in real-world field environments. Consequently, the presence of such errors and uncertainties makes it challenging to determine which sensor represents the true turbidity levels more accurately in field conditions. One potential positive of the low-cost sensor is its greater stability of readings during wet weather conditions. 

#### 3.3.3. Statistical Analysis of the Comparison between Our Low-Cost Sensor and the GreenSpan Sensor

There are statistically significant relationships between the low-cost and the GreenSpan turbidity sensor at both inlet (ρ = 0.69; *p* < 0.01) and outlet (ρ = 0.33; *p* < 0.01), demonstrating the performance of the low-cost novel sensor (Figure 5). The trend line slope at the inlet is 0.745, indicating that the majority of data points are clustered around the identity line, with a few points exhibiting higher turbidity readings from the low-cost sensor. However, the outlet’s slope of 0.22 indicates that more data points fall below the identity line, suggesting higher turbidity readings from the GreenSpan sensor compared to the low-cost sensor. As discussed above, the disparities were caused by both of the sensors’ inherent errors and the harsh field environments; they are also the reason for the weak correlation between the sensors’ results, especially for the outlet.

#### 3.3.4. Biofouling Impact of the Sensors

Biofouling was detected for both the low cost and the Greenspan sensors, at both the inlet and the outlet (Figure 6, top left). Biofouling was more prominent when testing the sensors in low turbidity levels (25th percentile), where the difference between before cleaning and after cleaning was sometimes greater than 50%. The observed high relative difference in turbidity readings may stem from the low background turbidity levels, where minor changes can appear more significant, thus showing a higher percentage change. This effect diminishes in conditions of higher turbidity (at the 50th and 75th percentiles). Furthermore, instances where after-cleaning turbidity readings are lower than before-cleaning (indicated by a positive *R_bio_*) can be attributed to the calibration procedure. During the calibration of low-cost sensors, the process begins with placing the “before-cleaning” sensor in the solution to take a reading, followed by measuring the solution’s turbidity with a handheld turbidity meter. The sensor is then removed for cleaning and reinserted into the same solution for an “after-cleaning” reading. Removing the “before-cleaning” sensor can dislodge dirt and biofilm, contaminating the solution and potentially increasing its true turbidity, increasing the sensor’s voltage drop and resulting in a modification to the calibration curve such that a positive *R_bio_* is estimated. Despite these calibration discrepancies in the low-cost sensors, the Wilcoxon Rank Sum test reveals no significant differences between the low-cost and the Greenspan sensors at both the inlet and the outlet locations, indicating that, statistically, the low-cost sensor and the GreenSpan sensor offer similar performances despite the variabilities observed.

Furthermore, the analysis of the relationship between the deployment time and the relative difference in turbidity readings, illustrated in Figure 6 (bottom left), showed no significant correlation. The data points scattered around the axis suggest that neither the low-cost nor the GreenSpan sensor had a detectable biofouling trend with time. This indicates that time of deployment is not necessarily an important factor in governing the degree of biofouling and that perhaps other factors (e.g., those that control algae growth, such nutrient levels and light exposure [69]) are more important for fouling of the sensors.

#### 3.3.5. Drift of the Sensors

The drift of the sensors was observed for both sensors at both the inlet and the outlet deployment locations (Figure 6, top right). Drift was more apparent for the low-cost sensors than for the GreenSpan (greater deviations from 0%; Figure 6, bottom right). Indeed, there was a significant difference detected between the drift values obtained by the GreenSpan and the low-cost sensor at the inlet. This could be due to the quality of the plastic cover used to protect the low-cost device, which may scratch more readily (as opposed to the glass on the GreenSpan) from high-velocity debris at the inlet and because of frequent cleaning. Some differences observed here could also be attributed to the contamination of the calibration solution [70]. Similarly to the biofouling issue results, no significant relationship was found between the relative difference in drift and the cumulative deployment period (Figure 6, bottom right), suggesting that sensor scratching is likely not cumulative and perhaps the occurrence is more episodic or acute in nature (e.g., a large scratch occurring once during maintenance, rather that the accumulation of smaller scratches over time).

## 4. Conclusions and Future Work

In conclusion, this paper presents the development and testing of a novel turbidity sensor, which has a low cost of around USD 23.50. The sensor’s operational current and sleep mode current were measured, demonstrating its energy efficiency. Both laboratory experiments and field tests were conducted to validate the sensor’s performance. The lab calibration established a linear relationship as the calibration curve for turbidity measurements. The field performance validation involved comparing the sensor’s time series data with those of the GreenSpan sensor, and the results showed that the low-cost sensor effectively detected turbidity changes in the water column. The linear regression statistical test was applied to analyse the sensor data, which showed differences between the data, but a statistically significant relationship was still observed between the sensor results. No correlation was detected between biofouling and the deployment time of the sensors; however, discrepancies in readings were observed, likely originating from the calibration procedure. Furthermore, in the biofouling analysis of both sensors, no significant biofouling was found on either the low-cost sensor or the GreenSpan sensor. This implies that the same maintenance requirements could be applied to the low-cost sensor as to the GreenSpan sensor. Additionally, drift in the low-cost sensor readings was noted, which could be attributed to scratches resulting from harsh environmental conditions and frequent cleaning; this problem is linked to the quality of the transparent epoxy board and could be improved by further hardware enhancements. Overall, the low-cost turbidity sensor demonstrates a potential opportunity to apply the sensor to capture high temporal and spatial data from complex water systems.

For future improvements, several modifications can be made to both the circuit design and the housing to enhance the sensor’s performance. Regarding the circuit design, the analogue and digital grounds can be decoupled to eliminate unnecessary noise from the circuit. Moreover, more algorithms will be compared in different numbers (10, 100, and 1000) of measurement within 1 s, which may provide a more energy-efficient way for turbidity monitoring. Additionally, incorporating a self-cleaning function could reduce the maintenance frequency required for the sensor. Furthermore, the sensor housing can be designed with different types of mounting methods to accommodate a wider range of field scenarios. By implementing these modifications, the turbidity sensor’s performance and its applicability in various field conditions can be significantly improved.

## Figures and Tables

**Figure 1 sensors-24-03926-f001:**
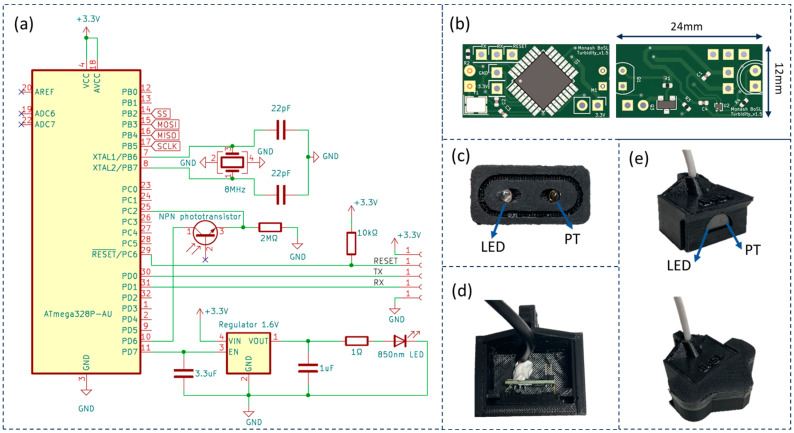
(**a**) circuit diagram of the innovative turbidity sensor; (**b**) PCB 3D view with dimension of the turbidity sensor (left for front side, right for back side); (**c**) 3D printed LED and phototransistor holder to arrange them at fixed positions, and each component has a 45 degree angle holder that makes the LED and phototransistor face each other with a 90 degree angle; (**d**) the top view of the PCB arrangement in the sensor housing, with the LED and phototransistor holder fixed in front of the housing and the PCB sitting in the middle of the sensor case; and (**e**) assembled turbidity sensor with a proper 3D printed housing, without cover assembled (**top**) and with cover assembled (**bottom**).

**Figure 2 sensors-24-03926-f002:**
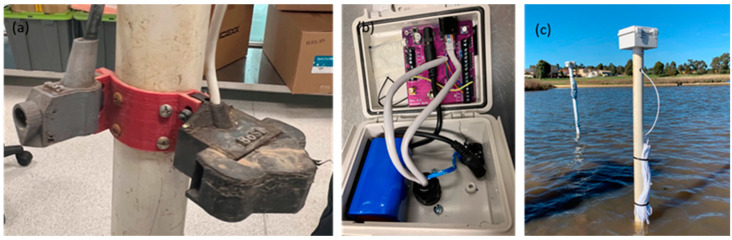
(**a**) low-cost sensors fixed on the PVC pipe; (**b**) low-cost sensor logger box; and (**c**) low-cost sensor package in the wetland.

**Figure 3 sensors-24-03926-f003:**
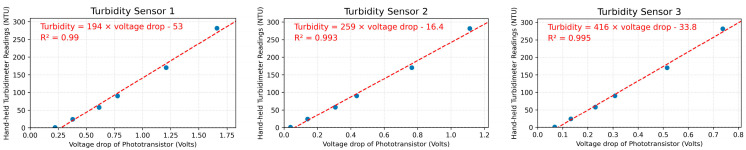
Regression of measured turbidity and the low-cost sensor’s outputs. The blue dots indicate the measuring results vs. the actual turbidity reading, the red dash line is the trend line of the best fits of the results, and the red words displayed in the graphs indicate the equation of the trendline and the R square value.

**Figure 4 sensors-24-03926-f004:**
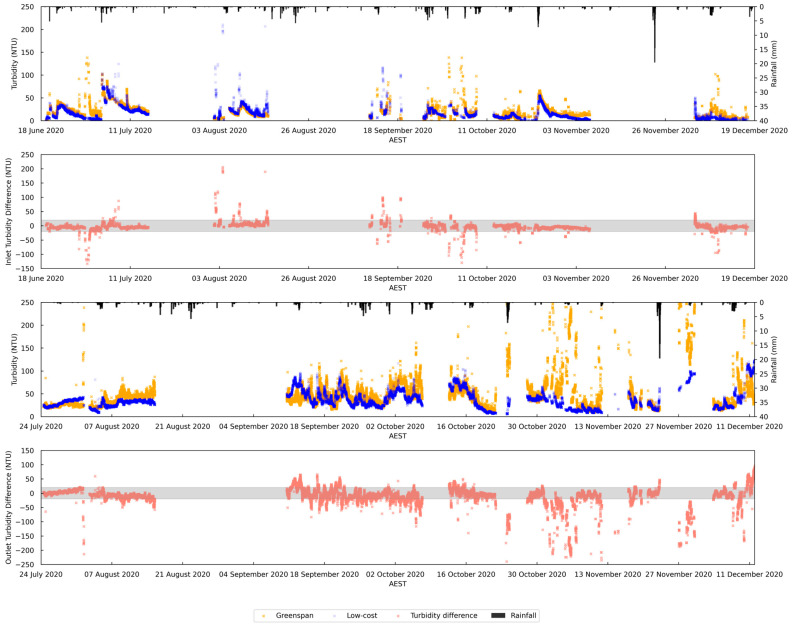
Time series data (after cleaning) and result difference between low-cost sensor and GreenSpan sensor. From the top to the bottom: the first and third subplots show the time series plot of the inlet data and outlet data, respectively; the second and the forth subplots show the turbidity difference (low-cost sensor readings minus Greenspan sensor readings) between both sensors for inlet and outlet, respectively; the positive values show greater low-cost sensor data and negative values show greater Greenspan sensor data; and the shaded area shows ±20 NTU difference, which is 97% of the total data at inlet and 73% at the outlet.

**Figure 5 sensors-24-03926-f005:**
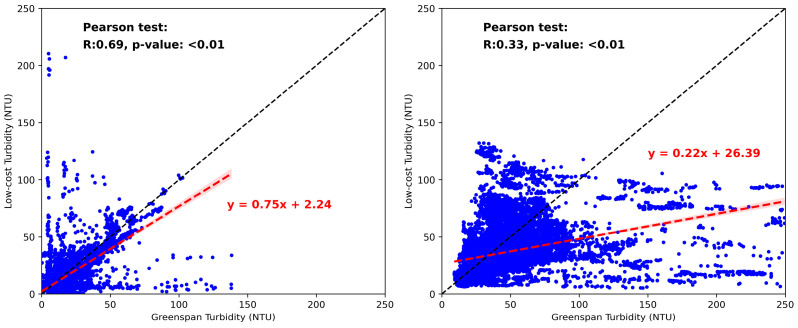
Plot of the GreenSpan sensor results vs the low-cost sensor results at both wetland inlet (**left**) and outlet (**right**). The black dash line shows the identity line, and the red dash line is the best fit line for the data series. The linear trendline equation is shown in red colour; for the inlet trendline, the estimated slope is 0.75 [95% CI: 0.73, 0.76], the estimated intercept is 2.24 [95% CI: 1.92, 2.57], and for the outlet trendline, the estimated slope is 0.22 [95% CI: 0.21, 0.23], and the estimated intercept is 2.24 [95% CI: 25.88, 26.89]. The R value and *p*-value of Pearson test are shown in the diagram respectively.

**Figure 6 sensors-24-03926-f006:**
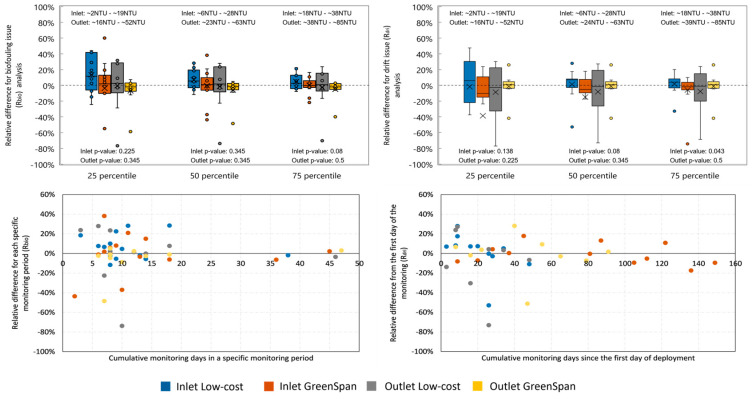
Plots of biofouling analysis and drift analysis. The left two graphs show the biofouling results, the top left graph is the boxplots of the relative difference (*R_bio_*) of biofouling effect, and the bottom left graph shows the relative difference (*R_bio_*) against the cumulative monitoring days in each specific monitoring period. Negative values mean the “after-cleaning” turbidity is smaller than the “before-cleaning” turbidity in a specific raw reading, and positive values mean the “after-cleaning” turbidity is greater than the “before-cleaning” turbidity in a specific raw reading. The right two graphs show the drift results, the top left graph is the boxplots of the relative difference of drift effect, and the top left graph shows the relative difference of the drift effect against the cumulative monitoring days since the sensor deployment. Negative values mean the “after-cleaning” turbidity before sensor is deployed is smaller than the “after-cleaning” turbidity after sensor is deployed in a specific raw reading, and positive values mean the “after-cleaning” turbidity before sensor is deployed is greater than the “after-cleaning” turbidity after sensor is deployed in a specific raw reading. Wilcoxon Rank Sum test results of comparing the box plots of low-cost sensor and GreenSpan sensor for both biofouling and drift effects are also shown below the box plots. Due to the axis limitation, some outliers are not shown in the boxplots (1 outlier for biofouling analysis 25 percentile, 1 outlier for drift analysis 25 percentile, and 1 outlier for drift analysis 50 percentile). The turbidity ranges for different percentile at both inlet and outlet are also presented on the top of the top graphs. For the plots of 25 percentile and 75 percentile in biofouling and drift effect against deployment period, please see Appendix B.

**Table 1 sensors-24-03926-t001:** Detailed cost of the low-cost turbidity sensor.

Parts	Cost in USD
LED	1.20
Phototransistor	3.20
PCB	15.00
Potting compound	1.00
Epoxy cover	0.10
3D-printing house	3.00
Price in total	23.50

**Table 2 sensors-24-03926-t002:** Results of lab power usage showing the measured current consumption in various modes, and the accumulated battery charge use over a year of operation using 1 min measurement intervals.

Mode	Time	Current	Battery Charge Use
Working Mode with active LED	1 s	88 mA	0.024 mAh
Working Mode with inactive LED	1 s	4 mA	0.0011 mAh
Sleeping Mode	58 s	<0.1 µA	<0.0001 µAh
Yearly Power use			13.43 Ah

**Table 3 sensors-24-03926-t003:** Percentage of the removed data based on different criterions. Data fitting multiple criteria has been counted under the first matching criterion. Therefore, no removed data have been counted under multiple columns in this table.

Sensor	Criterion 1: If in Water (%)	Criterion 2: Missing Data (%)	Criterion 3: Beyond Detecting Range (%)	Criterion 4: Continuous Trend Data (%)	Criterion 5: Dirt on Prob (%)	Criterion 6: Long Time after Maintenance (%)	Criterion 7: Erratic Gradients Values (%)	Total (%)
Inlet low-cost	0.0	3.9	20.9	8.5	0.0	20.8	9.6	63.6
Inlet GreenSpan	0.0	7.0	12.3	0.0	0.0	26.1	10.8	56.2
Outlet low-cost	0.0	5.8	3.4	3.6	0.0	32.3	12.8	57.8
Outlet GreenSpan	0.0	0.0	1.5	0.0	0.0	39.2	11.4	52.1

## Data Availability

The data presented in this study are available in the Appendix A.

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
