# Peer review of "A Compact, Low-Cost, and Low-Power Turbidity Sensor for Continuous In Situ Stormwater Monitoring"

_sensors, 2024, doi:10.3390/s24123926_

Round 1
Reviewer 1 Report
Comments and Suggestions for Authors
The paper describes a field trial of a low-cost in-situ turbidity meter. The work is overall good quality, relevant, and significant. However, the result could be made stronger by:
1) further describing the mechanical system/enclosure. Fig 1 describes the electrical components and shows images of the enclosure. It is not clear how the LED and detector are oriented in the 3d printed housing. Further, you could make these drawings available for others to use or improve.
2) it isn't clear why this particular LED and phototransistor were selected.
3) The sensor operation of taking 1000 measurements in 1 second (light/dark) seems arbitrary. Did you test 100 (or 10) measurements and find 1000 made a significant improvement?
4) You calibrate with 25, 50, 100, 150, and 250 NTU solutions. Does this cover the range of interest in your application?
5) You state ambient light is a concern but did you test in dark/light scenarios to ensure your approach is effective at mitigating this effect?
6) the final results (Fig 4 + appendix B), show significant differences, which are discussed. Can you quantify these differences in terms of NTU? For example, is the developed sensor within +/- 1,10,50 NTU of the reference sensor and under what conditions is this true?
Comments on the Quality of English LanguageSomething is wrong with the abstract, line 19, "results of the solution" seems out of place.
Author Response
Dear reiver,
Please see the attached attachment for our reply.
Thank you!

Reviewer 2 Report
Comments and Suggestions for Authors
Dear authors!
Thank you for the opportunity to get acquainted with the article you submitted! The article has a good structure. A lot of work has been done.
However, the following comments arose regarding the article.
1) The review needs improvement. Many citations from the reference list are mentioned in an integrated form (like [28-52]), which is unacceptable for a Q1 journal. A specific analysis of the content of previous studies on the topic is given only for a small set of papers.
2) Unfortunately, the operating principle of the new sensor does not contain scientific novelty and is a well-known measurement method. The paper presented is, therefore, more engineering than scientific. The selected main components of the measuring device are only listed; their choice is not supported by anything (any considerations, analysis of requirements, analysis of compliance with the required characteristics). The paper doesn't contain the main technical features of the core components - the reader is simply presented with the fact that this is exactly the range of electronic components used. The choice of microcontroller of the ATmega family also raises questions. The sufficiency of the proposed ADC bit depth (10 bits) has not been studied.
3) The scheme presented in Fig.1a also causes questions on circuit design (although I would like to emphasize again that this is an engineering work, not a scientific one). Is analogue and digital ground decoupling organized? If so, how? Why isn't C1 a polar capacitor? What is the current for Q1? Why not use the FET? The schematic from Fig.1a is ill-prepared (too many extra unusual angles in connection lines). Is PCB presented in Fig.1b developed by the authors? What is the information that the reader should find out from Fig. 1b?
4) The paper contains typos and design inaccuracies (for example, explanations of the expression (1) symbols should be written in italics).
5) The assessment of the cost of the sensor is purely for informational purposes and does not provide any special information. For Table 1, the time moment, for which the prices presented were relevant, was not even recorded. There are no links to support the price shown. This information has no scientific value.
6) Scientific novelty may lie in a description of the operating algorithms of the proposed sensor, but the paper does not make a comparison with existing analogues. So, it is hard to assess the novelty of this part of the presented paper.
7) Why the Pearson correlation test? The comparison and discussion of other approaches of linear association between the two datasets are necessary.
8) Fig.3 demonstrates that the given relationship is quadratic rather than linear (for all three sensors in the figure and most sensors from Appendix A). The quadratic approximation will pass the testing hypothesis on the degree of the approximating polynomial (the nonlinearity of the dependence and the non-randomness of the regression residuals if applying linear regression are visible). Why has the appropriate check not been carried out?
9) Fig.4 demonstrates that the cleaning procedure should be improved: we see a lot of unreliable points from the Greenspan sensor.
10) Paragraph 3.3.3 should contain confidence intervals for the value of the correlation coefficient. The correlation coefficient value of 0.33 indicates a weak relation (by Chaddock scale) between the sensors, which reduces the significance of the results presented in the paper. Probably, other metrics for comparing results are needed.
Comments on the Quality of English LanguageThe minor changes in English are necessary.
Author Response
Dear reviewers,
Please see the attachment for our reply.
Thank you!

Reviewer 3 Report
Comments and Suggestions for Authors
The author needs to briefly describe the technical specifications (accuracy, precision) of the Greenspan and handheld turbidimeters used as references. The authors need to provide more details about the technical specifications of the reference devices and the calibration procedures used. Additionally, using a standard reference calibrator would provide a more reliable basis for comparison.
Figure 3 shows the difference between the developed sensor and the reference, making it difficult to determine if the developed sensor is accurate. Why not use a standard reference that can be used as a calibrator?
Figures 4 and 5 significantly differ between the developed sensor and the reference device (Greenspan). Therefore, it is hard to conclude whether the developed device is working as expected or still needs improvement.
Author Response
Dear reviewer,
Please see the attachment for our reply.
Thank you!

Round 2
Reviewer 2 Report
Comments and Suggestions for Authors
Dear authors!
Thank you for the revising you have done to improve the article. I found answers to all the comments I made. I disagreed with a small number of them, but of course, this is not an obstacle to publishing the article.
I wish the authors good luck and further success in the future development of the sensor they presented!
Comments on the Quality of English LanguageThe quality of English is sufficient. Only few typos correction is necessary.